# Evaluation of the Effectiveness of Basic Palliative Care Training for Primary Care Nurses in a Health Area in Spain: A Quasi-Experimental Study

**DOI:** 10.3390/healthcare13192419

**Published:** 2025-09-24

**Authors:** Isidro García-Salvador, Encarna Chisbert-Alapont, Amparo Antonaya Campos, Clara Hurtado Navarro, Silvia Fernández Peris, Luis Alberto Gómez Royuela, Paz Rodríguez Castellano, Jorge Casaña Mohedo

**Affiliations:** 1Faculty of Nursing and Podiatry, University of Valencia, 46010 Valencia, Spain; isidro.gs@hotmail.com (I.G.-S.); hurtado_cla@gva.es (C.H.N.); 2Research Group INCUE, Valencia Health Department, Doctor Peset, 46017 Valencia, Spain; camposampa@gmail.com (A.A.C.); silviafp9@gmail.com (S.F.P.); pazisla13@gmail.com (P.R.C.); 3Nurse Hematology Service, Valencia Health Department La Fe, 46026 Valencia, Spain; 4Valencia Health Department, Doctor Peset, 46017 Valencia, Spain; 5Nurse Training Service, Valencia Health Department, Doctor Peset, 46017 Valencia, Spain; 6Carena Association of Psycho-Oncology, Valencia Health Department, Doctor Peset, 46017 Valencia, Spain; 7Nurse Primary Care Ruzafa Health Center, Valencia Health Department, Doctor Peset, 46017 Valencia, Spain; gomez_luiroy@gva.es; 8PDI Nursing Department, Faculty of Medicine and Health Sciences, Universidad Católica San Vicente Mártir, 46001 Valencia, Spain; jcasamo@gmail.com; 9PDI Faculty of Nursing, International University of Valencia, 46001 Valencia, Spain

**Keywords:** training, nursing, palliative care, primary care, practical skills

## Abstract

**Background/Objectives**: The general training in palliative care (PC) offered does not meet the needs of nurses and does not usually impact their clinical practice. The aim of the present study is to analyze the efficacy of a Palliative Care training plan, created and adapted to the specific needs of primary care nurses from the Department of Health Valencia, Doctor Peset. **Methods**: We executed the designed training plan offered by all the nurses in the department in five sessions lasting a total of 15 h through an active teaching methodology. A quasi-experimental pre-test/post-test study was conducted. The efficiency of the training provided was assessed through a self-administered, validated, anonymous questionnaire (INCUE instrument). Focus groups were conducted with the coordinators of the center to qualitatively assess the results and to propose lines of improvement. **Results**: The specific training provided to 85 nurses increased the application of PC in all areas of clinical practice (beginning of PC, communication skills, management of symptoms and care plans, legislation, bioethics at the end of life, and coping and loss). After the training, 88.8% passed the practical portion compared to 53.2% who did so previously. The area of lower impact was coping and loss or grief care. The coordinators perceived an improvement in palliative care, indicating the creation of a care protocol as a line of improvement. The percentage of nurses who felt sufficiently or very prepared to work with palliative patients practically doubled (from 23,5% to 42,4%). **Conclusions**: The directed training, based on the specific needs detected, was efficient and cost-effective. The methodology used had an impact on clinical practice.

## 1. Introduction

Primary care encompasses longitudinal care that ranges from the birth of a person to death and provides care within the family, which is the social and affective environment of an individual [1], caring for the individual, the family, and the community.

Palliative care (PC) focuses on providing care during the final stage of life, and its focus is aimed at improving the quality of life of the patient and the family. This type of care poses important challenges to achieve quality care in primary care, given that non-complex cases must be assumed [2]. On the one hand, the increasing demand is due to the increasing age of the population, and on the other, it is due to the scarce training of primary care nursing professionals on this subject, which is recognized and demanded by the nursing professionals themselves [3,4].

The scarce training on palliative care of nurses is a general problem [5,6,7,8,9] at the international level, due to the course load of this type of care in the training curricula [5,6,7,8,9,10], and in Spain, due to the unequal training provided, and at the university level, due to the heterogeneity and compulsory nature of the subject [8,11,12]. A total of 39.67% of national universities do not offer PC subjects, with a wide range of teaching loads (from three to six European Credits Transfer and Accumulation System). The Valencian Community has twelve nursing faculties, and only eight have the PC subject [12]. However, all the nurses, independently of their area of activity, must possess basic training in palliative care [13].

The European Association Palliative Care (EAPC) and the Spanish Association of Nursing in Palliative Care (AECPAL) have defined and recommended the basic knowledge or content needed for this training in the nursing degree [14,15].

Having knowledge does not necessarily mean having the necessary skills. Meaningful learning provides meaning to what is learned and integrates knowledge into daily clinical practice. The training must be focused on promoting skills and must be founded on the training needs of the professionals to whom it is directed [16]. On the other hand, it is necessary to assess if the participants who receive the training are applying these new skills so that they transfer what they have learned to their clinical practice [17].

The planning of a training program is a process that must include the identification of the training needs, the establishment of objectives, the learning outcomes, the teaching methodology, and the evaluation of the results and the efficacy of the training [18].

In a previous study on the training needs with respect to palliative care of primary care nurses from the Health Department Dr. Peset indicated that only 23.5% felt sufficiently or very prepared to provide care to palliative patients, and a similar percentage knew the care protocols at their center [19]. These same nurses perceived a need or a great need for training in 75.5% of the cases.

The assessment of these training needs in the Department was performed with the INCUE instrument [20]. This instrument evaluates PC knowledge and its application in clinical practice in the five areas described by the AECPAL for the basic-level training [21].

The results showed that many of the nurses possessed the knowledge (76.5%), but only a few applied it in their daily practice (18.6%), especially in the area of coping after loss and death (11.8%). Care for the bereaved is included in this area. This data is worrying, given that it implies that very little PC truly gets to the citizenry in the first level of care at the Health Department of Dr. Peset [19].

At the institutional and corporate level, measuring the efficacy of training or education is indispensable for management and business success. Measuring the effectiveness of education or training is found in ISO standard 9001:2015 [22].

Efficient training can be defined as training that achieves the objectives set, during the expected period of time, and at the estimated cost. Along the same line, the efficacy of the training process is evident in the degree to which the already-trained subjects put into practice the knowledge acquired. It must be noted that the implementation of the acquired knowledge into practice is not an intermediate step in its acquisition; it is necessary to give a prudent amount of time to be able to adequately measure its efficacy [23].

The main objective of the present study is to analyze the efficacy of a training plan for palliative care, created and adapted according to the specific needs of primary care nurses at the Valencian Health Department of Doctor Peset.

## 2. Materials and Methods

### 2.1. Training Plan

A group of experts designed a training plan that was specific and adapted to the needs detected and directed towards its practical applicability. It was based on a bibliographical search and the training on PC recommendations from the AECPAL [13]. Its creation was performed through two rounds of agreement, and the final document was subjected to an external review through the use of the Delphi method. In this plan, the professional profile to which it was directed was described, as well as its objectives, the education strategy, the teaching methodology, the syllabus, the course load, the profile of the professors, and the system of evaluation of the results. In addition, within the syllabus, in the area of coping and death, a protocol for caring for the bereaved was included, with follow-up and care guidelines [19].

An active teaching methodology was chosen, which converts students into the protagonists of their learning. With a face-to-face approach, problem-based learning and simulations were chosen, in which practical cases are presented and resolved, reflective learning is produced, and collaborative group work takes place. This type of approach achieves higher motivation and participation, and it allows contrasting the participants’ points of view and reasoning during each case. In addition, it promotes critical thinking and the evaluation of care plans developed in situations that could be faced during the professional career of the nurses [24]. This type of methodology is considered a tool that has an effect on changes in the practice in the fields of business, education, and healthcare [25].

The teaching profile of a nurse expert in palliative care and with clinical experience in home care allowed for the adequate development of the training sessions. The session about grief care also included the participation of a psychologist.

The timetable of the training sessions was established by the research team in coordination with the Nursing Department Management during normal working hours and with the least possible impact on the care of the population. The nurses were distributed into groups with a maximum of 25 people. The training sessions were repeated until they were completed by the 7 groups into which the population of the nurses from different primary care centers of the department was distributed. The coordinator attended all the training sessions.

The training plan consisted of 5 sessions lasting 3 h each. The sessions were programmed weekly on Tuesdays. The contents of the sessions were as follows:Identification of people in a palliative situation and complexity.Symptomatic monitoring and the subcutaneous route.Communication skills and difficult questions.Protocol and care for mourning.Last wishes and shared care planning.

The ethical aspects were addressed in a cross-cutting manner in the clinical situations posed and in the problem resolutions.

The training activity lasted 15 h. A priori, some authors indicate that only training lasting more than 150 h has an impact on evidence-based practice [26]. However, a class load of this magnitude has other drawbacks, such as the costs, the dropout rate, incomplete training, or lack of interest. It could also lead to a limited scope of the outcomes in the department due to the travel time that nurses are subjected to due to the competition for the consolidation of jobs. A shorter and more specific training is economically affordable for the management, as it maintains the motivation of the professionals and makes it possible to complete it.

The training was proposed for all the primary care nurses in the department, with a total duration of 12 months.

### 2.2. Design

A quasi-experimental study was proposed with a pre-test/post-test design, with the present study centered on the analysis of the post-training results. The answers of the group were compared with those obtained by the previous group, in which the training needs were assessed. The data from the pre-test participants were published beforehand [19].

The evaluation of the training was performed 6 months after the end of the training of all nurses.

The general criterion of the efficacy of the training was the absence of a statistically significant difference between passing the theoretical part and passing the practical part. The indicator used was the increase in the percentage of nurses who passed the theoretical part and the practical part with respect to the pre-training phase. All of this was justified by the good level of knowledge detected and its scarce practical application.

The evaluation of the barriers to clinical practice was performed through the extraction of the comments from the training session attendees and the focus group composed of the coordinators of the different health centers. The aim of the focus group was to obtain information about the opinions, attitudes, perceptions, and experiences related to the application of palliative care to the population after the training and the related changes or difficulties perceived.

### 2.3. Sample and Data Collection

The study population included primary care nurses of adults in the Valencia Health Department, Doctor Peset, who had received basic training on palliative care proposed by the department. The training was proposed to 173 nurses registered in primary care at the department, including 17 midwives and 35 pediatric nurses, with a total of 121 nurses in adult care.

The selection of the sample was non-probabilistic and intentional. The intention was to recruit all of the primary care nurses in the department who had received the training proposed by the department. The intentional non-probabilistic sampling was justified by the small population size. A random selection would imply an insufficient sample for obtaining statistically significant results. However, for a level of confidence of 95%, a precision of 3%, a ratio of 5%, and expected losses of 15%, a sample of at least 76 participants was considered necessary [27].

The subjects were recruited from the different primary care centers of the department. The coordinator was contacted to ask for their collaboration in the recruitment, given the importance of attaining the highest response rate possible among the nurses who had received the training. The department’s teaching assistant also collaborated. Participation was stimulated through weekly reminders via email or WhatsApp. The collection of data ended when no responses were obtained after 3 consecutive weeks or when a representative sample was obtained.

It must be considered that, as the training was proposed to all nurses in the department, some of them did not meet the inclusion criteria, as they were pediatric nurses or midwives. However, if they showed interest in the training, they were allowed to participate.

No reimbursement was offered to the subjects who declined to participate, nor was any type of promotional material encouraging or rewarding participation provided.

The inclusion criteria were as follows:Primary care nurses who exerted their professional activity with an adult population during the period of data collection, without the necessary professional experience in primary care.Having completed the basic training in palliative care offered by the department. Complete training was defined as attending all of the sessions, or missing one of the training sessions at most, which was verified through a signed record of session attendance.Providing a written consent for participation.

The exclusion criteria were as follows:
Nursing students.Nurses in training as residents.Nurses under contract whose initial provision was less than 30 days.Nurses who had not received the training offered or who missed more than one training session.Nurses who exclusively cared for a pediatric population and those who worked as midwives, given that they did not provide grief support or home visits.

The data were collected through an anonymous online questionnaire in Google Forms, from April to September 2024. The online self-report questionnaire contained a brief introduction, the objective of the study, inclusion criteria, the need for consent for participation, guaranteed anonymity, confidentiality, and the possibility of withdrawal. Each subject generated an identification code with which to request withdrawal.

The widespread access to the internet and the professional use of a single IP address allowed the entire strata of the study population to be represented, avoided self-selection biases, and ensured the safety of the participants. In addition, all the participants showed ratios similar to the total population with respect to sex, mean age, and training, according to the data provided by the statistical service of the department, which allowed considering the results as representative of the study population.

### 2.4. Instruments

The data collection instrument was the INCUE instrument [20]. This instrument is a self-administered questionnaire, designed ad hoc based on the bibliography, and it was previously used in a pilot study and validated. The instrument is the same one used for the assessment of the training needs of nurses. The training plan and the protocol for caring for the bereaved were designed based on these results.

The INCUE instrument assesses PC knowledge and its application in clinical practice by primary care nurses in 5 areas: PC principles, management of symptoms and care plans, coping with loss and death, communication skills, and ethical and legal aspects. The knowledge is assessed through questions with a yes/no dichotomous answer, and its practical application is measured by a 5-point Likert scale (never–always). The minimum score to pass the theoretical training is 18 points of the 23 possible correct answers, while for the practical training, it was 90 points out of the possible 120, with a 0 given to “never”, 1 to “rarely”, 2 to “sometimes”, 3 to “frequently”, and 4 to “always”.

The questionnaire contains some questions about training on palliative care and the perceived need for preparation on the subject; the training needs that are not covered (through multiple answers). It also includes a question about knowledge of the protocols of PC care at their workplace. Other sociodemographic and professional questions will describe the sample.

The estimated time for completing it was 15 to 20 min.

### 2.5. Data Analysis

Univariate and bivariate descriptive analyses were performed through the calculation of the mean and standard deviation for the quantitative variables and the frequency and ratios for the qualitative ones.

Bivariate comparison tests were also performed to complete the descriptive analysis. The Wilcoxon–Mann–Whitney or Kruskal–Wallis tests were used according to the number of categories to compare the quantitative variables, and the χ^2^ test with a simulation of the *p*-value with 2000 replicates was used to compare the qualitative variables.

Spearman’s correlation was calculated between the accumulated scores of the theoretical and practical block, pre- and post-training. This allowed comparing the relationship between the variables and provided additional information about the behavior of these dimensions in the different groups.

All the analyses were performed with R statistical software, version 4.2.2.

The databases were consolidated into a single set, integrating the observations of the different moments in the study, pre- and post-training.

The feedback results of the participants about the training sessions and the focus group of coordinators with respect to the barriers detected were analyzed by the research group. The topics were grouped as a function of their similarity into categories. The posterior triangulation through techniques such as the identification of recurring themes and contextual interpretations allowed for the identification of topics, patterns, and perspectives about the subject. Afterwards, the conclusions were extracted by consensus, as well as the more significant recommendations of the information collected [28].

### 2.6. Ethical Considerations

This study was reviewed and approved by the Pharmaceutical Research Ethics Committee of the Dr. Peset University Hospital (Research Project FEAPCP24-V01 and code CEIM 133/24).

The participants received a detailed description of the study and were guaranteed confidentiality and anonymity. The sociodemographic data solicited only allowed for the description of the sample, without the identification of the participant being possible. They were also informed about the voluntary nature of their participation and their withdrawal without negative consequences. In compliance with their right to withdraw from the study, each participant could generate an identification code that could help in their identification in case they requested the withdrawal of their answers from the study.

The signed informed consent from each participant was obtained.

The anonymity of the participants was maintained during both the pre- and post-test to favor participation and to decrease social desirability bias. Even though the pairing of the pre- and post-test answers from the same subject could enrich the results from a methodological perspective, as the training was proposed by the work institution where the professionals worked, they could have understood it as a possible form of monitoring.

The research team complied with the requirements set forth in the Organic Law on the Protection of Personal Data and the Guarantee of Digital Rights (3/2018 of 5 December), implementing the necessary measures.

The study was conducted in compliance with current ethical and legal standards (Declaration of Helsinki).

## 3. Results

### 3.1. Training

Of the 121 adult care nurses, 105 completed the training (86.77%). A total of 97 answers were received, with eight nurses who did not complete the training who were, therefore, excluded. The answers from one midwife and three pediatric nurses were also excluded, although they had completed the training, given that they did not meet the inclusion criteria. Ultimately, the answers from 85 nurses were included. The response rate was 80%.

The sociodemographic data are shown in Table 1. The group showed similar characteristics to the group of participants who took part in the detection of training or pre-training needs, with slight differences. Among them, we found a higher response rate among the male nurses and PC nurses.

Most of the participants (83.5%) had completed the training by attending all the sessions.

From the total, 82.4% considered the training to be sufficiently or very useful for clinical practice. The data also indicated a marked decrease in the need for training in all areas (Table 1). However, more than half of the participants considered that they needed more training in the management of symptoms, and about a third of them said that they needed more socio-familiar and psycho-emotional training, communication skills, or training on grief care. Nevertheless, the perception of needing more training (quite a lot and a lot) decreased from 75.5% to 43.5%.

The training also improved knowledge of the protocols for PC care in the workplace, from 22.5% to 55.3%.

The perception of being sufficiently or well prepared to work with patients who needed palliative care increased from 23.5% in the pre-training group to 42.4% in the post-training group (Table 1).

The mean scores, segmented according to the subjective perception of preparation for PC, increased in both the theoretical block and the practical block. The percentages of passing the theoretical and practical blocks also increased (Table 2). The differences in the groups with a lower perception of preparation were statistically significant.

The scores obtained by the post-training group were higher, in both the total score and according to blocks (theoretical and practical), and in the different areas, with a statistically significant difference between them (Table 3).

Spearman’s correlation between the accumulated scores of the theoretical and practical blocks before the training was 0.49, with a *p*-value < 0.001, while the scores in the post measurement were 0.26, with a *p*-value of 0.015. These data allow us to confirm that the training was efficient, as there were no statistically significant differences between the theory and the practice after the training.

It is important to emphasize that the answers from the pre-training group indicated that the training needs were practical in nature. In this previous assessment, the results indicated that the nurses had a theoretical capacity of 76.5% (passing the theoretical block of the questionnaire), but this training was not translated into practice. All of the post-training answers showed that they were trained at the theoretical level. However, improvement in the practical training was not reached by half of the group, although it doubled the results from the pre-training group.

In the results according to areas, the increase in all the percentages after the training must be underlined at both the theoretical and practical levels and with statistically significant differences in most of them. In the practical areas, the passing percentages were doubled (Table 3). The lowest results were found in the area of coping with loss and death (19.4%), which includes care for the bereaved. In these areas, the starting percentage of passing in the pre-training group was low (11.8%).

Table 4 shows the increase in the percentages of practical application in the activities assessed in the questionnaire. The frequent and consistent (always) use of some instrument to identify the patients with palliative needs increased from 34.3% in the pre-training group to 68.2% in the group that received training. As already indicated, the activities related to coping with loss and death obtained the lowest percentages in the putting into practice aspect, frequently and always, although in these activities, the lowest percentages were observed in the pre-training group in clinical practice. Thus, the periodic follow-up of family members after the death of a patient was only performed by 12.7%, increasing to 37.7% after the training, which included a specific protocol of care for the bereaved. Along the same line, we find that the use of tools to assess the risk of complicated grief began with a percentage of 10.7%, increasing to 34.1% after the training. The referrals to psychology/psychiatry for people considered at risk of complicated grief were frequently and always performed by 22.6% of the participants in the pre-training group, which increased to 44.1% after the training, which is in agreement with the guidelines of the bereaved care protocol and which outlined referral guidelines.

Another of the activities with a scarce clinical practice was the spiritual care as part of the care provided to palliative patients, which was only manifested by 6.9% of the answers before the training as an activity that was frequently and always performed. The answers by the post-training group indicated that 29.4% performed this activity in an equal manner.

In the area of ethical and legal aspects, the participation in the making of decisions improves its practical application, although it was found with low percentages. In the pre-training group, 25% indicated that they frequently and always participated, while in the post-training group, 38.8% did so.

The communication skills reached, at the practical level, high percentages in the frequent or consistent (always) application of the activities that were assessed, taking into account non-verbal language and exploring concerns and the feelings of the patient or family needs (Table 4).

### 3.2. Feedback or Barriers for the Application of Palliative Care

The results were presented to the coordinators of the health centers after their analysis. They considered that they fit the reality they perceived after the training and considered the training plan and the teaching methodology as being very adequate.

The training had improved the care of individuals with palliative needs, although they considered that better results had not been obtained due to other problems, such as a lack of resources or the need for a multidisciplinary approach and the involvement of other team members. This is in addition to the difficulty in identifying the patients due to a lack of coordination or communication between the levels of care (Table 5).

The care of the bereaved obtained a lower score in clinical practice, according to the coordinators, due to the difficulty in providing care in the emotional sphere. In addition, we find the fear that nurses could have when “bothering” the person in grief, and on the other hand, their respect with regard to pain that they could be feeling at that moment in time. They also believed that they needed care from all the members of the team, not only nursing.

For areas of improvement, they listed the training of other professionals on the team, the creation of specific care plans or guides, and computer algorithms that ease the care and indicate the procedural steps at all times.

During the training sessions, the nurses contributed with their difficulties and concerns, in line with those provided by the coordinators. In addition, some of the nurses indicated having difficulties with the care of people with palliative needs due to personal barriers related to recent, unresolved grief processes or negative experiences with the topic of death.

## 4. Discussion

The training proposal was completed by a high percentage of nurses, which was similar to or greater than that obtained by other authors with other proposals [29,30].

The interest or perception of the need to become trained in PC could have led to this result, but the programming within the workday could have also encouraged attendance. On the other hand, the duration of the training plan (15 h) could have contributed to and helped in its completion. Despite some authors believing that a longer training period guarantees the acquisition of skills [26], other training programs with a similar class load were also effective, although they only dealt with pain management [31]. In our training plan, a higher class load would not have been able to be integrated into the workday due to its high cost. On the other hand, it must be considered that professionals who take part in 150 h training programs seem to be motivated and interested in the acquisition of skills and do so voluntarily outside their work hours. In the current training program evaluated, the training was recommended to nurses based on the previously detected training needs.

The face-to-face teaching methodology is also a factor that could have impacted attendance and finishing the program. This training allows for a higher degree of development of communication skills and teamwork (very important factors in palliative care), as well as a faster resolution of doubts [32]. This type of experimental learning is preferred by the adult population and has a higher impact than online training [33].

We believe that the methodology based on problem resolution and real clinical situations had an effect on rating the usefulness of the training offered. Identifying situations that were experienced, or potential ones, helps in considering the training as useful. In addition, this type of learning has a stronger impact according to the literature, which ensures the existence of a link between emotion and learning [34,35].

Despite the training provided, a high number of participants believed that they needed more training, and this perception was lower at the beginning or compared to that detected in other studies [5,36,37]. It must be noted that the training plan is not only intended to cover the needs detected and provide training on basic skills, especially at the practical level.

On the other hand, it is logical to think that such a short period of training cannot cover all the training needs. In addition, putting into practice the knowledge acquired can pose new challenges in the care and interventions performed, which result in new training needs.

The new training needs encompass almost all of the dimensions of an individual, from the most physical, such as the management of symptoms, to those in the emotional, cognitive, social, or spiritual spheres. The latter were also demanded in other studies, in which the nurses believed that they must be trained in communication skills [5,36,38] or spiritual care [39]. Arantzamendi et al. found that Spanish nurses seemed to focus more on physical care and not as much on emotional, social, or spiritual health, perhaps due to this lack of preparation [4]. However, the positive interaction of these spheres contributes to the comprehensive development and the general well-being of the person, having an influence on the reduction in suffering and an increase in their quality of life, which are the aims of PC. The demand for training on spirituality could be justified by its specific absence of this content in the training plan.

Communication is the foundation of most of the care provided, being indispensable when making an assessment of the needs, and for health education or accompaniment. In PC, practical communication skills are needed, both verbal and non-verbal, to be able to understand the emotional environment of the patient and the family and to be able to interact with them [40,41]. This skill is also essential for working in groups, exerting leadership, or resolving conflicts.

In the training plan, a session was dedicated to communication and how to address the answers to difficult questions. Possibly, given the number of participants, it was not possible to cover all their needs or concerns.

Despite the efforts by the managers or those responsible for the health centers to increase awareness about the procedures related to PC, on many occasions, the nurses were not aware of them or did not use them regularly. Although the computerized standardized care plans are easy to use, they must be understood before they can be applied. The training plan used included a bereavement care protocol that specified assessment and recommended interventions during each visit, as well as their timetables. In addition, during the sessions, other existing protocols were alluded to, and information was provided on how to consult them. All of this supports an improvement in knowledge about the protocols.

The self-perception of the skills needed to provide care to individuals with palliative care needs and their families improved greatly after the training, although more than half of the nurses believed that they were not sufficiently prepared for it. The training helps with this self-perception [30,31,37,42,43], but we believe that it is not the only factor that has an impact on this perception. Contact with death and suffering needs preparation, but also a certain degree of awareness, predisposition, or motivation [40,44]. The attitudes of the nurses when facing death or end-of-life situations and experiences are another factor that has an influence on basic skills [42]. The participating nurses had an extensive professional career, although their experience was not specifically in PC or primary care.

The training provides the knowledge necessary to manage situations that occur in clinical practice and to provide the most adequate care, but a more humanistic type of knowledge is also needed [38]. Self-care and self-management of emotions that result from the continuous contact with suffering, meaningful work, and adequate support or workloads are elements necessary for the prevention of fatigue due to compassion [45].

Many authors support the need to develop specific programs to improve basic PC skills [16,42,46], such as the training plan, whose efficacy was assessed in the present study. However, in all of these previous studies, the assessment was based on the knowledge acquired after the training, on the self-efficacy or self-confidence, or on the attitudes when caring for dying patients, assuming that providing knowledge will have a positive repercussion on its application in clinical practice and the improvement of care [29,41,47].

The previous assessment of training needs at the theoretical level already showed a high level of knowledge, and all the participants passed the test after the training with at least the minimum score needed. The good results of this theoretical or knowledge training may have been due to the adaptation of the training plan to the training needs of the participants, as recommended by some authors [17,18,40]. However, in the same previous assessment and in other similar studies, it was observed that having the knowledge did not necessarily imply its application in practice [19]. Some authors already supported the need for an assessment of the application of newly learned skills [17].

The knowledge acquired after the training reached levels that were higher than after other training programs, although it must be taken into account that it is difficult to compare the results given that different instruments were used to measure them [36,47,48]. The efficacy of the training was measured based on the improvement of attitudes towards the care of patients with palliative care needs [33,46]. Other authors assessed the self-efficacy or self-confidence perceived after the training [30,37,43,49].

None of the reviewed articles that assessed training measured the practical application of the theoretical knowledge acquired. Our results showed a substantial improvement in the practical skills, observed as the increase in the activities that represented each of the training areas of the plan (Table 4). However, the putting into practice of the knowledge did not reach maximum levels. In some of the areas, such as coping with loss and death, it was lower than expected, as only a third of the participants passed this part, despite the efforts of providing a protocol that would facilitate this type of care and attention to the bereaved. It is conceivable that if better results had been obtained in this area and similar ones in the remaining areas, the number of participants who would have passed the practical portion would have been significantly higher.

In the remaining practical areas, the results of the application in clinical practice were significantly better, underlining the greater abilities in communication skills or management of ethical and legal aspects.

It may be that the requirement to frequently or always apply the knowledge in the activities assessed in each area could be too high a standard to consider it as the minimum required, but these activities are considered part of the basic nursing skills in palliative care.

The presence of professional or personal barriers is another factor that must be considered to justify the results obtained at the practical level, despite possessing the necessary knowledge.

With respect to the professional barriers that could make addressing PC difficult for primary care professionals, some authors have revealed difficulties in the communication between team members, alleviation of symptoms, or coordination with the community [5]. In our study, the difficulties observed were also related to problems in teamwork or the scarce participation of nurses in decision-making. These obstacles have already been mentioned in other studies, in which nurses felt excluded in both the development of protocols and in the decision-making processes, while the perception of doctors was that the decisions were made jointly [50].

Another of the professional barriers described was the high workloads. It must be noted that the ratio of nurses per 1000 inhabitants in Spain is 6.3, as compared to 8.5 in the European Union, so this barrier could have a certain justification [51]. However, this should not serve as an argument for the lack of care of individuals in a palliative situation.

On the other hand, personal barriers, such as recent unresolved grief or negative experiences, can condition the attitude towards the care of people who need palliative care or people in mourning. After the pandemic ended, health personnel in general were affected in their mental health to a greater or lesser degree, affecting the morbidity of primary care personnel in a striking manner [52]. Based on this, we must keep in mind that some of the nurses may have their own emotional problems related to this event but who do not feel able to face a new situation of emotional stress that could increase their discomfort.

The lack of motivation is a difficult factor to address, given that it can present a never-ending number of causes and manifestations. Exploring the causes individually could provide improvement proposals, but it is a difficult challenge to overcome. Even then, we believe that our training proposal, which was initiated by the management and is easy to attend during work hours, can be an incentive for participating, which could create an interest in the topic and improve the attitude of professionals towards this type of care.

As for the teacher profile, different recommendations exist. Some studies support inter-professional education due to its collaborative learning strategy, its effectiveness in healthcare, and the greater acquisition of knowledge, skills, and attitudes of the students [53]. However, others indicate that this type of education does not have an impact on professional practice; it is difficult to establish general recommendations [54]. We believe that a nursing teaching profile can help in the motivation for the practical application and in promoting change, given that nurses can see themselves mirrored as equals, as shown by other authors [55].

The training plan evaluated has a considerable effectiveness on the practical application of the knowledge learned, as shown by the results. On the other hand, although the purpose of the study was not to assess cost-effectiveness, the timetable of a weekly session and the adaptation of care schedules accordingly allowed for maintaining community care without the need to increase human resources during the training period. On the other hand, the teachers were freed from their care duties during the training sessions. Only one teacher needed funds for travel and stay, as a profile necessary for the training was not found in our department. No teachers received financial compensation for their time, given that it was during the workday. In addition, audiovisual means and physical spaces were owned by the department and could be used without an additional cost. The total cost for the training was less than EUR one thousand.

We believe that if the desire is to change palliative care, just as other authors have argued [42], clinical nursing leaders must define personalized strategies and interventions based on the needs of the nurses in their teams. In this way, specific deficits or factors can be addressed, and the continuous development of the nurses’ skills in palliative care can be promoted. For this, institutions must be proactive in the training, and we must not wait until the nurses show interest in the training to solicit it or to enroll in existing courses. Training plans tailored to each area of action and previously identified needs must be proposed, and the importance of the training must be emphasized as part of their job performance. The efficacy demonstrated with the training plan of the present study fits this second option and is ideal for achieving improvements in palliative care.

The present study has many limitations. In first place, the data analyzed were self-declared by the participants. For this, there could be a certain degree of influence of social desirability and a degree of exaggeration of the perception of skills in the clinical practices of PC applied, despite the guarantee of anonymity of the participants to minimize this bias. In second place, the results or practical impact of the training were assessed from the perspective of nursing, which is the profession that provides the care, and not from the perspective of the people in a palliative situation, who are the recipients of this care. Thus, the practical application of PC was measured indirectly. Furthermore, the type of sampling used could lead to self-selection bias, despite measures implemented to minimize it.

In the future, it would be interesting to reproduce the study in other areas of primary care to observe if similar results can be obtained. In addition, exploring barriers or difficulties that interfere with the emotional or grief care may be useful, as these obtained the worst results in the practical aspect. On the other hand, validated instruments must be developed that would allow us to discover the clinical impact of palliative care training performed. This would allow us to assess, aside from its efficacy, the improvement of the training in later editions.

## 5. Conclusions

The PC training based on the specific needs of primary care nurses from a health department was efficient, having an impact on the clinical practice of care activities performed by these nurses.

A face-to-face teaching methodology based on problem resolution seems to have an effect on the modification of attitudes and practical skills, and we believe that this methodology can be recommended in nurse training with a low class load. In addition, it allows for interaction with the teachers, who are able to explore the difficulties or barriers of the participants to address them as much as possible. Nevertheless, the assessment of the training needs must include the identification of said difficulties or barriers to adapt the design of the training to them.

## Figures and Tables

**Table 1 healthcare-13-02419-t001:** Descriptive data of the sociodemographic characteristics and training needs by group.

	Group	
Variables	Before N = 102	After N = 85
**Age (years), mean (SD)**	48.29 (11.92)	45.79 (11.80)
**Sex, n (%)**		
Men	10 (9.8%)	15 (17.6%)
Women	92 (90.2%)	70 (82.4%)
**Maximum level of professional qualification, n (%)**		
Diploma/degree	72 (70.6%)	56 (65.9%)
PhD	2 (2.0%)	3 (3.5%)
Nurse or specialist	19 (18.6%)	17 (20.0%)
Master’s	9 (8.8%)	9 (10.6%)
**Type of job, n (%)**		
Coordinator of the center	11 (10.8%)	6 (7.1%)
Nurse	80 (78.4%)	69 (81.2%)
Pediatric nurse	8 (7.9%)	6 (7.1%)
Palliative care nurse	1 (1.0%)	3 (3.5%)
Community case manager	2 (2.0%)	1 (1.2%)
**Professional experience (years), mean (SD)**	21.17 (12.43)	19.56 (11.38)
**PC training at Dr. Peset, n (%)**		
Yes, I have completed the full training, attending all sessions.	0 (NA%)	71 (83.5%)
Yes, I have completed the training partially, missing one training session	0 (NA%)	14 (16.5%)
**Usefulness in the clinical practice, n (%)**		
None	0 (NA%)	1 (1.2%)
Little	0 (NA%)	4 (4.7%)
Some	0 (NA%)	10 (11.8%)
A lot	0 (NA%)	30 (35.3%)
Very	0 (NA%)	40 (47.1%)
**Level of PC training n (%)**		
Basic (25–80 h)	57 (55.9%)	34 (40.0%)
Intermediate (80–150 h)	18 (17.6%)	18 (21.2%)
Advanced (Master’s or PhD)	3 (2.9%)	4 (4.7%)
I have not received training	24 (23.5%)	29 (34.1%)
**Do you perceive a need for more training, n (%)**		
None	1 (1.0%)	4 (4.7%)
Little	1 (1.0%)	2 (2.4%)
Somewhat	23 (22.5%)	42 (49.4%)
Quite a bit	58 (56.9%)	34 (40.0%)
A lot	19 (18.6%)	3 (3.5%)
**Preparation perceived for working with PC patients, n (%)**		
None	4 (3.9%)	2 (2.4%)
Little	26 (25.5%)	8 (9.4%)
Somewhat	48 (47.1%)	39 (45.9%)
Quite a bit	22 (21.6%)	35 (41.2%)
Very prepared	2 (2.0%)	1 (1.2%)
**Knowledge of care protocols in PC at the workplace, n (%)**		
No	58 (56.9%)	29 (34.1%)
Does not know/no answer	21 (20.6%)	9 (10.6%)
Yes	23 (22.5%)	47 (55.3%)
**Need for training in**		
Symptomatic control, n (%)	71 (69.6%)	51 (60.0%)
Socio-family matters, n (%)	41 (40.2%)	24 (28.2%)
Communication skills, n (%)	53 (52.0%)	27 (31.8%)
Grief and coping with loss, n (%)	67 (65.7%)	28 (32.9%)
Spirituality, n (%)	25 (24.5%)	14 (16.5%)
Palliative care principles, n (%)	40 (39.2%)	13 (15.3%)
Ethical aspects, n (%)	42 (41.2%)	16 (18.8%)
Need for psycho-emotional training, n (%)	67 (65.7%)	32 (37.6%)
Other areas, n (%)	4 (3.9%)	6 (7.1%)

**Table 2 healthcare-13-02419-t002:** Descriptive scores and achievement segmented by subjective preparation according to group.

	Not at All–Somewhat Prepared			Quite–Very Prepared		
Group	Before N = 78	After N = 49	*p*-Value ^1^	Before N = 24	After N = 36	*p*-Value ^1^
**Variables**						
**Total score**			<0.001			0.057
Mean (SD)	83.28 (21.65)	107.14 (17.59)		103.42 (19.80)	113.92 (13.39)	
Minimum, maximum	27.00, 119.00	57.00, 137.00		55.00, 132.00	89.00, 140.00	
**Pass blocks of theory and practice, n (%)**			0.001			0.6
No	69 (88.5%)	30 (61.2%)		14 (58.3%)	18 (50.0%)	
Yes	9 (11.5%)	19 (38.8%)		10 (41.7%)	18 (50.0%)	
**Theoretical score**			<0.001			0.028
Mean (SD)	18.87 (2.67)	21.41 (1.50)		20.38 (2.45)	21.69 (1.51)	
Minimum, maximum	12.00, 23.00	18.00, 23.00		14.00, 23.00	18.00, 23.00	
**Pass the theory block, n (%)**			<0.001			0.060
No	21 (26.9%)	0 (0.0%)		3 (12.5%)	0 (0.0%)	
Yes	57 (73.1%)	49 (100.0%)		21 (87.5%)	36 (100.0%)	
**Practical score**			<0.001			0.063
Mean (SD)	64.41 (20.18)	85.73 (17.43)		83.04 (18.81)	92.22 (12.64)	
Minimum, maximum	14.00, 98.00	35.00, 115.00		37.00, 113.00	67.00, 117.00	
**Pass practice block, n (%)**			<0.001			0.6
No	69 (88.5%)	30 (61.2%)		14 (58.3%)	18 (50.0%)	
Yes	9 (11.5%)	19 (38.8%)		10 (41.7%)	18 (50.0%)	
**Practical score (clustered), n (%)**			<0.001			0.061
14–44	16 (20.5%)	1 (2.0%)		1 (4.2%)	0 (0.0%)	
45–59	18 (23.1%)	1 (2.0%)		2 (8.3%)	0 (0.0%)	
60–74	16 (20.5%)	12 (24.5%)		3 (12.5%)	3 (8.3%)	
75–89	19 (24.4%)	16 (32.7%)		8 (33.3%)	15 (41.7%)	
90–105	9 (11.5%)	11 (22.4%)		9 (37.5%)	9 (25.0%)	
106–117	0 (0.0%)	8 (16.3%)		1 (4.2%)	9 (25.0%)	

^1^ Wilcoxon–Mann–Whitney rank-sum test/Pearson chi-square test with simulated *p*-value (2000 replicates). Abbreviations: standard deviation (SD).

**Table 3 healthcare-13-02419-t003:** Descriptive score classifications by area according to group.

	Group		
Variables	Before N = 102	After N = 85	*p*-Value ^1^
**Passes theory and practice blocks, n (%)**			<0.001
No	83 (81.4%)	48 (56.5%)	
Yes	19 (18.6%)	37 (43.5%)	
**Total score (SD)**	88.02 (22.81)	110.01 (16.21)	<0.001
**Passes theory block, n (%)**			<0.001
No	24 (23.5%)	0 (0.0%)	
Yes	78 (76.5%)	85 (100.0%)	
**Theoretical score (SD)**	19.23 (2.69)	21.53 (1.50)	<0.001
**Passes practice block, n (%)**			<0.001
No	83 (81.4%)	48 (56.5%)	
Yes	19 (18.6%)	37 (43.5%)	
**Practical score (SD)**	68.79 (21.31)	88.48 (15.83)	<0.001
**Passes the area of PC principles (theory), n (%)**			0.13
**No**	4 (3.9%)	0 (0.0%)	
Yes	98 (96.1%)	85 (100.0%)	
**Score in the area of PC principles (theory), mean (SD)**	3.64 (0.56)	3.96 (0.19)	<0.001
**Passes the area of symptom management and care plans (theory), n (%)**			<0.001
No	38 (37.3%)	8 (9.4%)	
Yes	64 (62.7%)	77 (90.6%)	
**Score in the area of symptom management and care plans (theory)**	3.67 (1.02)	4.40 (0.73)	<0.001
**Passes the area of coping with loss and death (theory), n (%)**			<0.001
No	24 (23.5%)	1 (1.2%)	
Yes	78 (76.5%)	84 (98.8%)	
**Score in the area of coping with loss and death (theory), mean (SD)**	4.16 (0.94)	4.73 (0.47)	<0.001
**Passes the area of communication skills (theory), n (%)**			0.039
No	16 (15.7%)	5 (5.9%)	
Yes	86 (84.3%)	80 (94.1%)	
**Score in the area of communication skills (theory), mean (SD)**	3.29 (0.92)	3.60 (0.60)	0.032
**Passes the area of ethical and legal aspects (theory), n (%)**			0.013
No	11 (10.8%)	1 (1.2%)	
Yes	91 (89.2%)	84 (98.8%)	
**Score in the area of ethical and legal aspects (theory), mean (SD)**	4.47 (0.69)	4.84 (0.40)	<0.001
**Passes the area of PC principles (practice), n (%)**			<0.001
No	79 (77.5%)	36 (42.4%)	
Yes	23 (22.5%)	49 (57.6%)	
**Score in the area of PC principles (practice), mean (SD)**	14.21 (4.43)	17.56 (3.73)	<0.001
**Passes the area of symptom management and care plans (practice), n (%)**			<0.001
No	75 (73.5%)	33 (38.8%)	
Yes	27 (26.5%)	52 (61.2%)	
**Score in the area of symptom management and care plans (practice), mean (SD)**	13.12 (5.48)	18.20 (4.12)	<0.001
**Passes the area of coping with loss and death (practice), n (%)**			0.002
No	90 (88.2%)	60 (70.6%)	
Yes	12 (11.8%)	25 (29.4%)	
**Score in the area of coping with loss and death (practice), mean (SD)**	9.79 (5.83)	14.52 (4.71)	<0.001
**Passes the area of communication skills (practice), n (%)**			<0.001
No	64 (62.7%)	25 (29.4%)	
Yes	38 (37.3%)	60 (70.6%)	
**Score in the area of communication skills (practice), mean (SD)**	16.07 (4.22)	19.28 (3.27)	<0.001
**Passes the area of ethical and legal aspects (practice), n (%)**			<0.001
No	68 (66.7%)	26 (30.6%)	
Yes	34 (33.3%)	59 (69.4%)	
**Score in the area of ethical and legal aspects (practice), mean (SD)**	15.61 (3.95)	18.92 (3.17)	<0.001

^1^ Wilcoxon–Mann–Whitney rank-sum test/Pearson chi-square test with simulated *p*-value (2000 replicates). Abbreviations: standard deviation (SD).

**Table 4 healthcare-13-02419-t004:** Descriptive statements of the questions in the practical part of the INCUE questionnaire according to the group.

Area	Variables/Questions	Group		*p*-Value ^1^
		Before N = 102	After N = 85	
**Principles of PC**				
**You work as a team in your healthcare activity, n (%)**				0.2
	Never	0 (0.0%)	2 (2.4%)	
	Rarely	6 (5.9%)	2 (2.4%)	
	Sometimes	13 (12.7%)	10 (11.8%)	
	Frequently	51 (50.0%)	35 (41.2%)	
	Always	32 (31.4%)	36 (42.4%)	
**You assess the needs of family members, n (%)**				0.071
	Never	1 (1.0%)	1 (1.2%)	
	Rarely	6 (5.9%)	1 (1.2%)	
	Sometimes	18 (17.6%)	7 (8.2%)	
	Frequently	47 (46.1%)	39 (45.9%)	
	Always	30 (29.4%)	37 (43.5%)	
**You intervene in the needs of family members, n (%)**				0.004
	Never	2 (2.0%)	1 (1.2%)	
	Rarely	7 (6.9%)	2 (2.4%)	
	Sometimes	42 (41.2%)	19 (22.4%)	
	Frequently	38 (37.3%)	35 (41.2%)	
	Always	13 (12.7%)	28 (32.9%)	
**You use some instrument to identify patients with palliative needs, n (%)**				<0.001
	Never	19 (18.6%)	1 (1.2%)	
	Rarely	23 (22.5%)	10 (11.8%)	
	Sometimes	25 (24.5%)	16 (18.8%)	
	Frequently	26 (25.5%)	42 (49.4%)	
	Always	9 (8.8%)	16 (18.8%)	
**You provide spiritual care as part of the care of the palliative patients you care for, n (%)**				<0.001
	Never	36 (35.3%)	11 (12.9%)	
	Rarely	31 (30.4%)	19 (22.4%)	
	Sometimes	28 (27.5%)	30 (35.3%)	
	Frequently	5 (4.9%)	21 (24.7%)	
	Always	2 (2.0%)	4 (4.7%)	
**You assess the quality of life of palliative patients, n (%)**				<0.001
	Never	6 (5.9%)	1 (1.2%)	
	Rarely	13 (12.7%)	1 (1.2%)	
	Sometimes	19 (18.6%)	6 (7.1%)	
	Frequently	28 (27.5%)	26 (30.6%)	
	Always	36 (35.3%)	51 (60.0%)	
**Management of symptoms and care plans**				
**You use some type of rating scale in your daily work, n (%)**				<0.001
	Never	3 (2.9%)	1 (1.2%)	
	Rarely	11 (10.8%)	1 (1.2%)	
	Sometimes	47 (46.1%)	15 (17.6%)	
	Frequently	36 (35.3%)	44 (51.8%)	
	Always	5 (4.9%)	24 (28.2%)	
**You perform or teach family oral care in dependent patients, with palliative sedation or short-term prognosis, n (%)**				<0.001
	Never	10 (9.8%)	1 (1.2%)	
	Rarely	19 (18.6%)	6 (7.1%)	
	Sometimes	24 (23.5%)	10 (11.8%)	
	Frequently	34 (33.3%)	37 (43.5%)	
	Always	15 (14.7%)	31 (36.5%)	
**After administering a prescribed pain control drug at home, you assess its effectiveness, n (%)**				<0.001
	Never	6 (5.9%)	1 (1.2%)	
	Rarely	20 (19.6%)	4 (4.7%)	
	Sometimes	21 (20.6%)	5 (5.9%)	
	Frequently	30 (29.4%)	28 (32.9%)	
	Always	25 (24.5%)	47 (55.3%)	
**You teach the family how to prepare and administer subcutaneous medication, n (%)**				<0.001
	Never	18 (17.6%)	5 (5.9%)	
	Rarely	11 (10.8%)	10 (11.8%)	
	Sometimes	40 (39.2%)	14 (16.5%)	
	Frequently	18 (17.6%)	23 (27.1%)	
	Always	15 (14.7%)	33 (38.8%)	
**You use non-pharmacological measures to help with symptom control, n (%)**				<0.001
	Never	18 (17.6%)	2 (2.4%)	
	Rarely	16 (15.7%)	5 (5.9%)	
	Sometimes	32 (31.4%)	22 (25.9%)	
	Frequently	31 (30.4%)	40 (47.1%)	
	Always	5 (4.9%)	16 (18.8%)	
**You perform a periodic follow-up according to the needs of people in palliative situations, n (%)**				<0.001
	Never	15 (14.7%)	1 (1.2%)	
	Rarely	14 (13.7%)	3 (3.5%)	
	Sometimes	25 (24.5%)	9 (10.6%)	
	Frequently	30 (29.4%)	40 (47.1%)	
	Always	18 (17.6%)	32 (37.6%)	
**Coping with loss and death**				
**You identify the phases of coping with the disease of a palliative patient, n (%)**				<0.001
	Never	10 (9.8%)	3 (3.5%)	
	Rarely	20 (19.6%)	6 (7.1%)	
	Sometimes	37 (36.3%)	25 (29.4%)	
	Frequently	31 (30.4%)	35 (41.2%)	
	Always	4 (3.9%)	16 (18.8%)	
**You facilitate the expression of emotions by family members after the patient’s death, n (%)**				0.001
	Never	4 (3.9%)	1 (1.2%)	
	Rarely	6 (5.9%)	5 (5.9%)	
	Sometimes	32 (31.4%)	8 (9.4%)	
	Frequently	34 (33.3%)	31 (36.5%)	
	Always	26 (25.5%)	40 (47.1%)	
**You perform a periodic follow-up with family members after the patient’s death, n (%)**				<0.001
	Never	40 (39.2%)	7 (8.2%)	
	Rarely	24 (23.5%)	14 (16.5%)	
	Sometimes	25 (24.5%)	32 (37.6%)	
	Frequently	10 (9.8%)	27 (31.8%)	
	Always	3 (2.9%)	5 (5.9%)	
**You assess the mourner’s support network, n (%)**				<0.001
	Never	30 (29.4%)	1 (1.2%)	
	Rarely	24 (23.5%)	9 (10.6%)	
	Sometimes	15 (14.7%)	27 (31.8%)	
	Frequently	24 (23.5%)	40 (47.1%)	
	Always	9 (8.8%)	8 (9.4%)	
**You use instruments to assess the risk of complicated grief, n (%)**				<0.001
	Never	43 (42.2%)	8 (9.4%)	
	Rarely	23 (22.5%)	22 (25.9%)	
	Sometimes	25 (24.5%)	26 (30.6%)	
	Frequently	8 (7.8%)	20 (23.5%)	
	Always	3 (2.9%)	9 (10.6%)	
**You refer people who are considered at risk of complicated grief to psychology/psychiatry, n (%)**				<0.001
	Never	41 (40.2%)	10 (11.8%)	
	Rarely	19 (18.6%)	21 (24.7%)	
	Sometimes	19 (18.6%)	19 (22.4%)	
	Frequently	16 (15.7%)	28 (32.9%)	
	Always	7 (6.9%)	7 (8.2%)	
**Communication skills**				
**You intervene in situations where the patient expresses discomfort or anger, n (%)**				<0.001
	Never	5 (4.9%)	2 (2.4%)	
	Rarely	22 (21.6%)	9 (10.6%)	
	Sometimes	41 (40.2%)	25 (29.4%)	
	Frequently	30 (29.4%)	32 (37.6%)	
	Always	4 (3.9%)	17 (20.0%)	
**For patients in a state of palliative sedation or with a decreased level of consciousness, the care to be provided is explained in advance, n (%)**				<0.001
	Never	15 (14.7%)	3 (3.5%)	
	Rarely	18 (17.6%)	11 (12.9%)	
	Sometimes	20 (19.6%)	7 (8.2%)	
	Frequently	32 (31.4%)	32 (37.6%)	
	Always	17 (16.7%)	32 (37.6%)	
**In your daily work, you take into account the patient’s non-verbal language, n (%)**				0.007
	Never	0 (0.0%)	0 (0.0%)	
	Rarely	1 (1.0%)	1 (1.2%)	
	Sometimes	11 (10.8%)	5 (5.9%)	
	Frequently	48 (47.1%)	23 (27.1%)	
	Always	42 (41.2%)	56 (65.9%)	
**You stand at the same height as the patients when conducting the interview for their assessment, n (%)**				0.002
	Never	2 (2.0%)	0 (0.0%)	
	Rarely	3 (2.9%)	0 (0.0%)	
	Sometimes	18 (17.6%)	6 (7.1%)	
	Frequently	46 (45.1%)	30 (35.3%)	
	Always	33 (32.4%)	49 (57.6%)	
**You explore the patient’s concerns and feelings, n (%)**				<0.001
	Never	1 (1.0%)	0 (0.0%)	
	Rarely	5 (4.9%)	0 (0.0%)	
	Sometimes	17 (16.7%)	6 (7.1%)	
	Frequently	52 (51.0%)	33 (38.8%)	
	Always	27 (26.5%)	46 (54.1%)	
**You identify the needs of the family, n (%)**				<0.001
	Never	1 (1.0%)	0 (0.0%)	
	Rarely	14 (13.7%)	0 (0.0%)	
	Sometimes	35 (34.3%)	16 (18.8%)	
	Frequently	32 (31.4%)	38 (44.7%)	
	Always	20 (19.6%)	31 (36.5%)	
**Ethical and legal aspects**				
**You tailor patient care to their preferences, n (%)**				<0.001
	Never	5 (4.9%)	1 (1.2%)	
	Rarely	8 (7.8%)	0 (0.0%)	
	Sometimes	31 (30.4%)	6 (7.1%)	
	Frequently	43 (42.2%)	41 (48.2%)	
	Always	15 (14.7%)	37 (43.5%)	
**You involve the patient and family in decision-making regarding care, n (%)**				<0.001
	Never	1 (1.0%)	0 (0.0%)	
	Rarely	3 (2.9%)	0 (0.0%)	
	Sometimes	17 (16.7%)	4 (4.7%)	
	Frequently			
	Always			
**You inform patients in palliative care that there is an advance directive or prior instructions document, n (%)**				<0.001
	Never	25 (24.5%)	4 (4.7%)	
	Rarely	17 (16.7%)	7 (8.2%)	
	Sometimes	20 (19.6%)	22 (25.9%)	
	Frequently	24 (23.5%)	28 (32.9%)	
	Always	16 (15.7%)	24 (28.2%)	
**You respect the patient’s decisions even if you do not consider them appropriate, n (%)**				0.2
	Never	0 (0.0%)	0 (0.0%)	
	Rarely	0 (0.0%)	0 (0.0%)	
	Sometimes	7 (6.9%)	5 (5.9%)	
	Frequently	42 (41.2%)	24 (28.2%)	
	Always	53 (52.0%)	56 (65.9%)	
**You participate in decision-making, n (%)**				<0.001
	Never	14 (13.7%)	5 (5.9%)	
	Rarely	25 (24.5%)	4 (4.7%)	
	Sometimes	37 (36.3%)	43 (50.6%)	
	Frequently	22 (21.6%)	24 (28.2%)	
	Always	4 (3.9%)	9 (10.6%)	
**You take into account the cultural characteristics of the person and/or family when providing care, n (%)**				0.003
	Never	2 (2.0%)	0 (0.0%)	
	Rarely	7 (6.9%)	2 (2.4%)	
	Sometimes	21 (20.6%)	7 (8.2%)	
	Frequently	40 (39.2%)	29 (34.1%)	
	Always	32 (31.4%)	47 (55.3%)	

^1^ Wilcoxon–Mann–Whitney rank-sum test/Pearson chi-square test with simulated *p*-value (2000 replicates).

**Table 5 healthcare-13-02419-t005:** Feedback or barriers to the application of palliative care.

Topics	Patterns	Perspectives
Resources and institutional structure for palliative care	Lack of human resources	“My problem is time” “We don’t have psychologists at the centers”
	Lack of structural resources	“I have a hard time identifying people in palliative care despite the tools” “We should have specific computerized protocols and decision-making algorithms that would facilitate the implementation of the care plan”
	Coordination/communication between levels of care	“We have family members who aren’t in our area, and we can’t follow up”
Teamwork	Multidisciplinary approach Training other team members	“Well, this is fine, but what about the doctors?” “PC and bereavement care require team care” “The other day I ran a complexity assessment, and it showed that it wasn’t a complex situation. But we’re not able to improve that patient’s quality of life”
Difficulty in caring for the emotional sphere	Insecurity Lack of training	“I’ve never done grief counseling” “There are unclear guidelines for primary care” “If we touch on emotions…ugh” “The emotional sphere is more difficult to address” “I’m afraid to be alone with that family member” “After the patient’s death, I have the feeling that I’m going to upset the family member or increase their suffering by reminding them of what they’ve experienced”
Personal resources	Problems related to the topic of death Previous experiences	“I’d rather not do this. My mother died a year ago” “After the death of a loved one, the nurse showed very little empathy, and I have unpleasant memories of the situation”
	Lack of motivation	“I want to say that I was forced to come here” “You chose a bad time to take the course. In a few months, half the department will be leaving” “If you like the subject, you already have the knowledge”
	Need for more training	“It’s a difficult area” “Outcomes have improved, but it doesn’t qualify you to care for patients in need of palliative care”

## Data Availability

The data from the study are those included in this manuscript.

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
