# Peer review of "Evaluation of the Effectiveness of Basic Palliative Care Training for Primary Care Nurses in a Health Area in Spain: A Quasi-Experimental Study"

_healthcare, 2025, doi:10.3390/healthcare13192419_

Round 1

Reviewer 1 Report

Comments and Suggestions for Authors

This manuscript presents a quasi-experimental study evaluating the outcomes of a training project aimed at developing palliative care skills among the nursing population of a Valencian Health Department (primary care). The growing population requiring palliative care, outside of specialized palliative care teams, where most needs can be identified, represents an emerging challenge for healthcare systems. Nurses are a key figure and point of reference for palliative care services and could play a crucial role in assessing needs and, consequently, providing accurate responses to those needs. I appreciated the article and the underlying project, but I have a few comments to make about the manuscript.

  1. As a reader, I would like to know if there are differences in post-training outcomes (15 hours of training) between nurses with prior palliative care training (if any nurses have prior palliative care experience/skills) and those without such experience/skills.
  2. As noted above, I sincerely appreciate the effort to provide basic palliative care training to nurses working in primary care, but I doubt that 15 hours of training can provide practical skills (basic knowledge may require less effort to internalize) to a nurse trained to work in other settings. The authors address this issue, pointing out that some authors [ref. 26] believed that only a 150-hour course could provide sufficient expertise in evidence-based practice. I found the arguments (sustainability and risk of dropout) too weak to justify reducing the course hours to 15 (10%).
  3. Is the post-training evaluation (i.e., the questionnaire) capable of assessing the practical skills developed during the 15-hour course? As the authors correctly point out, developing subject knowledge (PC) does not imply the ability to apply it in daily practice. Is a self-administered questionnaire an appropriate tool for independently assessing practical skills/abilities? For example, how does the questionnaire assess the ability to identify patients in need of palliative care? Through a clinical case or a self-assessment question?
  4. In training session 3.1, I would like to point out that, although most of the nurses involved in the training activities provided answers to the questionnaire, considering the 121 adult care nurses, the researchers collected responses from 70% (85 nurses) of the original sample. Did the authors consider any selection bias and do they have information on nurses who did not provide answers/did not meet the criteria (attending a sufficient number of training hours, etc.)?
  5. The discussion section seems too detailed. I would suggest reducing it.

Author Response

Dear Reviewer1,

We welcome all your comments and suggestions. We hope we've addressed them all and are sure they will enrich our work. We remain at your disposal for any further assistance you may have.

Comment 1: As a reader, I would like to know if there are differences in post-training outcomes (15 hours of training) between nurses with prior palliative care training (if any nurses have prior palliative care experience/skills) and those without such experience/skills.

Response 1: In the assessment of training needs (previously published https://doi.org/10.3390/nursrep13020078), nurses with a higher level of training obtained better results, with the correlation between the scores of both parts being positive and statistically significant depending on the level of training of the respondents.

The questionnaire questions addressed basic aspects of PC, and it was observed that the more advanced training they had, the better their scores, both theoretically and practically. The highest scores on the questionnaire, were obtained by subjects with postgraduate or non-university PC training.

These data already showed that prior training positively impacts both knowledge and practical application, so we did not consider analysing this difference post-training. On the other hand, the percentages of pre-training are similar in the pre- and post-training groups, with the majority having prior training at a basic or higher level.

Comment 2: As noted above, I sincerely appreciate the effort to provide basic palliative care training to nurses working in primary care, but I doubt that 15 hours of training can provide practical skills (basic knowledge may require less effort to internalise) to a nurse trained to work in other settings. The authors address this issue, pointing out that some authors [ref. 26] believed that only a 150-hour course could provide sufficient expertise in evidence-based practice. I found the arguments (sustainability and risk of dropout) too weak to justify reducing the course hours to 15 (10%).

Response 2: We understand that training lasting only 10% of the recommended duration may be insufficient and difficult to justify and it may not provide enough training to improve the clinical practice. However, this is the backbone of our study: to demonstrate that shorter training, tailored to the specific training needs of a group, is effective. The results indicate a significant improvement in clinical practice, with the percentage of successful conclusion of the practical portion of the questionnaire doubling in all areas assessed (Table 3).

Regarding the defense of this type of training and its sustainability, several factors must be taken into account. First, it was not requested by the nurses who received it, but rather proposed by the primary care management. Furthermore, it was provided free of charge and during working hours, taking into account the maintenance of adequate care for citizens.

Training all nurses in the health department required the allocation of human and material resources for a year, so extending the training duration to 150 hours would be unsustainable. Furthermore, a longer period would entail greater staff mobility, absences due to retirement or illness, etc., and therefore a greater number of nurses with incomplete training.

On the other hand, as we indicated in the limitations, we believe this type of training should continue to be studied.

Comment 3: Is the post-training evaluation (i.e., the questionnaire) capable of assessing the practical skills developed during the 15-hour course? As the authors correctly point out, developing subject knowledge (PC) does not imply the ability to apply it in daily practice. Is a self-administered questionnaire an appropriate tool for independently assessing practical skills/abilities? For example, how does the questionnaire assess the ability to identify patients in need of palliative care? Through a clinical case or a self-assessment question?

Response 3: Indeed, a self-administered questionnaire may not be considered an independent source of information on the practical application of PC to the population, given that it provides data from the perspective of the care provider, not the receiver. However, most studies on training evaluate it in the same way as ours; the results obtained are considered valid.

The INCUE questionnaire was specifically designed to assess PC knowledge, but also its practical application. It also includes a specific question about the respondent's use of an instrument to identify patients with palliative needs.

The questions asked whether the instrument is used never, rarely, sometimes, frequently, or always. Furthermore, in the section analysing difficulties or barriers to the practical application of PC, other organisational or management aspects emerged for identifying patients who cannot be addressed through training.

Comment 4: In training session 3.1, I would like to point out that, although most of the nurses involved in the training activities provided answers to the questionnaire, considering the 121 adult care nurses, the researchers collected responses from 70% (85 nurses) of the original sample. Did the authors consider any selection bias and do they have information on nurses who did not provide answers/did not meet the criteria (attending a sufficient number of training hours, etc.)?

The quasi-experimental design implies a lack of sample randomisation, which raises the possibility of selection bias. However, given the small population size and high response rate, along with the online data collection strategy, we believe this bias is controlled or minimised. This is justified by a sample that contains subjects from all strata of the population and is therefore representative of the population.

We do not have information on the nurses who did not wish to participate in the training. The responses of nurses who did not meet the inclusion criteria (midwives and pediatric nurses) were similar to those of the sample. It was not considered appropriate to include their responses in the analysis, given that they were not included in the pre-training analysis.

Comment 5: The discussion section seems too detailed. I would suggest reducing it.

Response 5: We appreciate the suggestion, but we felt it was important to compare our results with other studies in several aspects. Precisely because of the significant differences in training duration, design, and assessment of the impact it has on practical application. Nevertheless, we have modified it following your recommendation.

Reviewer 2 Report

Comments and Suggestions for Authors

Thank you very much for inviting me to review the manuscript titled “Evaluation of the effectiveness of basic palliative care training for primary care nurses in a health area in Spain.”  The article is interesting and provides some important findings for the research community and policymakers, especially in an ageing society. However, there are some major concerns for the authors’ consideration.

  1. Title: Informative. However, the title indirectly indicates it is an experimental study. However, the exact study type, such as quasi or comparative, must be presented in the title for better transparency.
  2. Abstract is fine, and the authors mentioned improvement. However, it is unclear unless the authors clearly specify the numbers, changes, significant status, etc, to create interest for the readers.
  3. I appreciate the authors for making a good attempt at making a contextual framework for this study in the introduction. However, strong justifications (rationale/novelty) are highly recommended by emphasizing specific local gaps (e.g., heterogeneity in Spanish PC training, lack of applied skill transfer).
  4. Furthermore, the introduction contains some repetitive content, which could be trimmed
  5. Methods: The authors explained the training plan well, and using a validated instrument is highly appreciated and commendable. However, the study design (quasi) needs to be mentioned explicitly in the limitations.
  6. Again, non-probability sampling could lead to bias.
  7. The authors mentioned non-parametric tests applied to the study data. Did the authors apply any specific tests to the data distribution?
  8. Also, how did the authors apply and find the improvement? More clarity is required.
  9. Moreover, it is unclear whether it is a quasi-experimental study or a mixed-method study. Because the authors mentioned something about feedback and grouping them, etc. The qualitative component is underdeveloped and lacks a proper explanation.
  10. Results are clear. Some areas, such as “coping with the loss,” require further analysis, as they are the lowest areas.
  11. Discussion is fair. But the authors overstate the discussion about the improvements (Please note that the study has a lot of limitations.
  12. The authors claimed that it was cost-effective. Is it speculative?
  13. Please revise the conclusion considering your study limitations.

Author Response

Dear Rviewer 2,

We appreciate all your comments and suggestions. We hope we've addressed all of them and are sure they will enrich our work. We've modified the tables to improve them and make them clearer.

We remain at your disposal for any further assistance you may have.

Comment 1: Title: Informative. However, the title indirectly indicates it is an experimental study. However, the exact study type, such as quasi or comparative, must be presented in the title for better transparency.

Response 1: We appreciate your suggestion and understand that some titles indicate the type of study used. However, we believe the proposed title reflects our work better.

Comment 2: Abstract is fine, and the authors mentioned improvement. However, it is unclear unless the authors clearly specify the numbers, changes, significant status, etc, to create interest for the readers.

Response 2: Following your recommendation, we have added some additional information to the abstract, trying to respect the journal's guidelines.

Comment 3: I appreciate the authors for making a good attempt at making a contextual framework for this study in the introduction. However, strong justifications (rationale/novelty) are highly recommended by emphasizing specific local gaps (e.g., heterogeneity in Spanish PC training, lack of applied skill transfer).

Response 3: We have added further justification in the introduction.

Comment 4: Furthermore, the introduction contains some repetitive content, which could be trimmed

Response 4: We have revised the introduction, removing some of its content.

Comment 5: Methods: The authors explained the training plan well, and using a validated instrument is highly appreciated and commendable. However, the study design (quasi) needs to be mentioned explicitly in the limitations.

Comment 6: Again, non-probability sampling could lead to bias.

Response 5 and 6: We add to the limitations the possible bias due to design and sampling.

Comment 7: The authors mentioned non-parametric tests applied to the study data. Did the authors apply any specific tests to the data distribution?

Response 7: Non-parametric tests were applied directly, without performing any additional tests to assess normal distribution conformity or homogeneity of variance. In small sample sizes, hypothesis tests to assess normality and homogeneity of variance, have very low power, while in large sample sizes, they have excessive power, detecting entirely irrelevant differences. Although nonparametric tests are slightly less powerful than parametric tests when the assumptions are met, they generally perform well in scenarios where they are not met, making them a suitable default option. Additionally, and perhaps most importantly, it doesn't make much sense to calculate a mean for variables without a metric.

Comment 8: Also, how did the authors apply and find the improvement? More clarity is required.

Response 8: In section 2.2 Design we specify:

“The general criterion of the efficacy of the training was the absence of a statistically significant difference between passing the theoretical part and passing the practical part. The indicator used was the increase in the percentage of nurses who passed the theoretical part and the practical part with respect to the pre-training phase. All of this was justified by the good level of knowledge detected, and its scarce practical application.”

The results show that the post-training scores did not show a statistically significant difference between theory and practice (p-value 0.015). This criteria was established to demonstrate the effectiveness of the training. Furthermore, practical training improved in all areas assessed.

Comment 9: Moreover, it is unclear whether it is a quasi-experimental study or a mixed-method study. Because the authors mentioned something about feedback and grouping them, etc. The qualitative component is underdeveloped and lacks a proper explanation.

Response 9: The results presented correspond to a quasi-experimental study, consistent with the study's objective. Based on the results obtained, we considered it useful to include in the manuscript the feedback or barriers obtained during the training sessions and the focus group. This part of the study is not related to the aim, as it does not assess the effectiveness of the training plan, but rather provides information on the difficulties encountered in the practical application of acquired knowledge. Furthermore, some of these barriers are personal and cannot be overcomed through training. Understanding them can help address them appropriately through other resources.

Comment 10: Results are clear. Some areas, such as “coping with the loss,” require further analysis, as they are the lowest areas.

Response 10: In Table 4, we have attempted to reflect the results regarding the practical application of knowledge in more detail. The table indicates the activities assessed in each area, with three of the six activities highlighted in the text.

Comment 11: Discussion is fair. But the authors overstate the discussion about the improvements (Please note that the study has a lot of limitations.

Response 11: We have taken your comments into account and have modified the discussion by moderating our considerations.

Comment 12: The authors claimed that it was cost-effective. Is it speculative?

Response 12: The direct financial cost of the training was €10 per nurse, resulting in improved theoretical and practical skills. We believe this figure and the results obtained allow us to affirm that the training was cost-effective.

Comment 13: Please revise the conclusion considering your study limitations.

Response 13: We have taken your comments into account and have modified our conclusions by moderating our considerations.

Round 2

Reviewer 1 Report

Comments and Suggestions for Authors

I do not have any further comments/suggestions for Authors

Author Response

Comment 1: 

I do not have any further comments/suggestions for Authors

Response 1: 

We appreciate all your suggestions that have helped us improve our manuscript.

Reviewer 2 Report

Comments and Suggestions for Authors

Dear Authors,
Thanks for making great efforts in revising the manuscript as per the comments.
However, it is not mandatory to include the words “quasi-experimental” in the title, but it is highly recommended for clarity and transparency.
You may see that most of the guidelines (including TREND) suggest that the information on how units were allocated (point 1) must be presented. I could not find the word quasi-experimental even in the abstract. It decreases the transparency. 
Wish you all the best.

Author Response

Comment 1: It is not mandatory to include the words “quasi-experimental” in the title, but it is highly recommended for clarity and transparency.

Response 1: We have finally considered your suggestion to be appropriate and have modified the title.

Comment 2: You may see that most of the guidelines (including TREND) suggest that the information on how units were allocated (point 1) must be presented. I could not find the word quasi-experimental even in the abstract. It decreases the transparency. 

Response 2: We have modified the abstract to specify the type of study design, as per your suggestion.